# Genome-Wide Analysis Elucidates the Roles of *AhLBD* Genes in Different Abiotic Stresses and Growth and Development Stages in the Peanut (*Arachis hypogea* L.)

**DOI:** 10.3390/ijms251910561

**Published:** 2024-09-30

**Authors:** Cuicui Wu, Baoguo Hou, Rilian Wu, Liuliu Yang, Gang Lan, Zhi Xia, Cairong Cao, Zhuanxia Pan, Beibei Lv, Pengbo Li

**Affiliations:** Institute of Cotton Research, Shanxi Agricultural University, Yuncheng 044000, China; houbaoguo04830807@163.com (B.H.); wurilian@163.com (R.W.); yll289@126.com (L.Y.); langang@sxau.edu.cn (G.L.); xiazhi19711104@sina.com (Z.X.); caocair@126.com (C.C.); panzhuanxia@126.com (Z.P.); mhslyub@126.com (B.L.)

**Keywords:** *Arachis hypogea* L., *LBD* gene family, expression pattern, salt stress

## Abstract

The lateral organ boundaries domain (*LBD*) genes, as the plant-specific transcription factor family, play a crucial role in controlling plant architecture and stress tolerance. However, the functions of *AhLBD* genes in the peanut plant (*Arachis hypogea* L.) remain unclear. In this study, 73 *AhLBDs* were identified in the peanut plant and divided into three groups by phylogenetic tree analysis. Gene structure and conserved protein motif analysis supported the evolutionary conservation of *AhLBDs*. Tandem and segment duplications contributed to the expansion of *AhLBDs*. The evolutionary relationship analysis of *LBD* gene family between *A. hypogaea* and four other species indicated that the peanut plant had a close relationship with the soybean plant. *AhLBDs* played a very important role in response to growth and development as well as abiotic stress. Furthermore, gene expression profiling and real-time quantitative qRT-PCR analysis showed that *AhLBD16*, *AhLBD33*, *AhLBD67*, and *AhLBD72* were candidate genes for salt stress, while *AhLBD24*, *AhLBD33*, *AhLBD35*, *AhLBD52*, *AhLBD67*, and *AhLBD71* were candidate genes for drought stress. Our subcellular localization experiment revealed that *AhLBD24*, *AhLBD33*, *AhLBD67*, and *AhLBD71* were located in the nucleus. Heterologous overexpression of *AhLBD33* and *AhLBD67* in *Arabidopsis* significantly enhanced tolerance to salt stress. Our results provide a theoretical basis and candidate genes for studying the molecular mechanism for abiotic stress in the peanut plant.

## 1. Introduction

The peanut plant (*A. hypogaea* L.) belongs to the genus peanut of the subfamily papilionidae of *Leguminosae*, which is an allotetraploid crop (2n = 4x = 40) [1,2]. Cultivated peanut is a major oilseed crop and a cash crop that is widely cultivated worldwide [3]. China is one of the largest peanut producers in the world, with a planting area of more than 4.6 million hm^2^ and a pod yield of more than 17 million tons, accounting for 40% of the total global output. This ranks China as the first in the world for peanut production [4]. As one of the main oil crops in China, peanut plays an increasingly prominent role in developing agricultural production and increasing the supply of agricultural products. According to statistics, the area of saline–alkali land, arid land, and semi-arid land in China exceeds 30 million hm^2^, accounting for a fourth of China’s cultivated land. However, less than 20% of the land has been reclaimed and utilized [5]. Therefore, improving the land utilization rate of saline–alkali land, arid land, and semi-arid land has positive effects on promoting agricultural development and ensuring the safety of important agricultural products in China. With the rapid development of contemporary biotechnology, high-throughput sequencing technology has become increasingly sophisticated and perfected. The complete genome sequencing of peanut diploid ancestors [6] and tetraploid cultivars [3,7,8] has been completed one after another, which provides an important foundation for the further mining of stress resistance-related functional genes in the peanut plant.

Transcription factors (TFs) play a very important role in a plant’s developmental processes and environmental stress responses by regulating downstream gene expression [9]. The Lateral Organ Boundaries Domain (*LBD*) gene family, also called the *AS2/LOB*, is one class of plant-specific transcription factors [10]. LBD proteins are comprised of a relatively conserved N-terminal region and a variable C-terminal region [11]. The N-terminal region includes an LOB domain that comprises a zinc finger-like motif (CX2CX6CX3C) for DNA-binding activity, a GAS block (Gly-Ala-Ser) sequence, and a leucine-zipper-like coiled-coil motif (LX6LX3LX6L), which presumably participate in protein dimerization [12,13]. The LOB structural domain and the variable C terminus together form the expression structure of LBD genes [14]. According to the specific protein sequences of the LOB domain in the N-terminus, the LBD family can be divided into two classes (Class I and Class II). Class I members consist of conserved zinc finger-like motifs, GAS blocks, and leucine-zipper-like coiled-coil motifs, while Class II members only have leucine-zipper-like domains [12,15,16].

The first *LBD* gene was identified in Arabidopsis [12], and *LBDs* were initially proved to regulate plant growth and development processes [17,18,19]. However, extensive studies have shown that *LBDs* also play an important role in hormone response [20,21,22], metabolic regulation [23,24], disease resistance [25], and heading stage [26], etc. Arabidopsis, as a model plant, has received the most in-depth studies on the functions of *LBD* genes, and at least 25 *AtLBD* genes have been cloned and functionally characterized [27,28,29,30]. Previous studies have shown that *AtLBD13*, *AtLBD16*, and *AtLBD29* are related with root development [30,31]. Silencing of the fusarium oxysporum susceptibility gene *AtLBD20* has been observed to enhance resistance to the pathogen [32]. *AtLBD29* has been observed to participate in auxin signaling, which inhibits fiber wall thickening [33]. Loss of function in *AtLBD36* has been found to result in shorter references and the downward curving of lateral organs such as flowers [34]. *AtLBD37* has been observed to be involved in anthocyanin biosynthesis and metabolic regulation [35]. Alongside this, *OsIG1* has been observed to participate in the development of floral organs and megagametophytes in rice [36], while *OsLBD37* has been found to be involved in the regulation of nitrogen metabolism [37]. Moreover, the wheat *LBD* gene *TaMOR* can increase root architecture and increase yield in crop plants [38]. Some studies have also reported that *LBD* genes are involved in responses to different abiotic stresses. In soybean, the expression of *GmLBD12* was found to be induced by drought, salt, cold weather, and hormones [39]. In potato, the expression of *StLBD2-6* and *StLBD3-5* in the stem was found to be induced under drought stress [40].

Until now, studies have been conducted to detect the *LBD* gene family in other plants, such as the 44 *LBD* genes found in maize; 31 in rice; 34 in barley; 47 in *Pinus massoniana*; 55 in Moso bamboo; and 53 in sweet potato plants, respectively [41,42,43,44,45,46]. At present, the systematic investigation of the *LBD* gene family has not yet been performed in the peanut plant. To better understand the dynamics of *LBD* gene evolution in peanut plants, we used bioinformatic methods to identify the *AhLBD* family genes. The evolutionary relationship, chromosome distribution, gene structure, conserved motifs, and collinearity relationship with other species were also performed for *AhLBD*s. Furthermore, the functional diversification, expression pattern in various tissues, and different responses to salt and drought stresses were systematically analyzed. Our study is essential to understanding the evolution and diversity of peanut *LBD* genes and their potential role in plant growth and stress responses. It provides new insights to help improve our understanding of the the function and regulation of homologous genes in the peanut plant’s development, and it also provides a foundation for future functional research.

## 2. Results

### 2.1. Identification of LBD Family Genes in Peanut

A total of 73 *LBD* family genes were named from *AhLBD1* to *AhLBD73* based on their chromosomal positions in *A. hypogaea*. All 73 genes encoded proteins ranging from 137 (*AhLBD66*) to 918 (*AhLBD1*) amino acids. The Predicted protein isoelectric points (pI) of *AhLBDs* ranged from 4.78 (*AhLBD60*) to 9.34 (*AhLBD72*), and their molecular weights (MWs) ranged from 15.05 kD (*AhLBD66*) to 102.02 kD (*AhLBD1*). The hydrophilicity of all AhLBD proteins ranged from −0.73 (*AhLBD29*) to −0.05 (*AhLBD50*). Additionally, all *AhLBD* proteins were predicted to be located in the nucleus (see Appendix A).

### 2.2. Phylogenetic Analysis of AhLBDs in Peanut and Arabidopsis

To investigate the evolutionary connection between the *LBD* gene family of *A. hypogaea* and *A. thaliana*, we reconstructed a phylogenetic tree of *LBD* protein sequences (see Figure 1). A total of 73 *AhLBDs* and 43 *AtLBDs* were classified into three groups based on the phylogenetic tree results and named Group I, II, III, respectively. There were 6 *AhLBDs* and 18 *AtLBDs* in Group I. Group II contained 18 *AhLBDs* and 6 *AtLBDs*. While 49 *AhLBDs* and 19 *AtLBD*s were divided into Group III. Except for the first group, in which the number of *AtLBDs* in Arabidopsis was greater than that of *AhLBDs* in the peanut plant, the number of *AtLBDs* in the other two groups was lower than that of *AhLBDs*.

### 2.3. Chromosomal Distribution and Gene Duplication Analysis of AhLBDs

Based on the genome databases data, chromosomal maps of *AhLBDs* were also constructed (see Figure 2). The *AhLBD* genes were distributed on all chromosomes of *A. hypogaea*. The distribution of *AhLBDs* in the A subgenome (36 genes) was similar from those in the B subgenome (37 genes). The maximum number of *AhLBD* genes was 6 in chr07 and chr17. A total of 5 *AhLBDs* were distributed on chr01, chr14 and chr18, 6 chromosomes contained 4 *AhLBD* genes, and 6 chromosomes contained 2 *AhLBDs*, respectively. Only chr08 contained 3 *AhLBD* genes. Among these, there were only 2 tandem duplication gene pairs, named *AhLBD1–AhLBD2* in chr01, and *AhLBD43–AhLBD44 in* chr12, which were substantially close to each other on their chromosomes, and they were also clustered together on the phylogenetic tree.

By collinearity analysis, there were 78 duplicate gene pairs within *AhLBD*s, which were distributed on different chromosomes (see Figure 3), indicating the occurrence of tandem duplication and segment duplication during the expansion of *AhLBD* genes.

### 2.4. Gene Structure and Conserved Protein Motifs Analysis of AhLBDs

To better understand the evolutionary relationships of *AhLBD* family genes, the conserved protein motifs and gene structure of *AhLBDs* were analyzed (see Figure 4). Three classes were divided according to the phylogenetic tree of *AhLBDs*. Class I contained the most *AhLBD* genes (53), Class II contained 14 *AhLBDs* while there were only 6 *AhLBDs* in Class III.

The conserved motif analysis of AhLBD protein sequences was performed using the MEME online tool, and 10 conserved motifs were predicted. It was found that different AhLBD proteins had certain differences in the number and distribution of motifs (see Figure 4A,B). Each gene contained 3 to 7 motifs, and all AhLBD proteins contained conserved LOB domains Motif2 and Motif3. The N-terminus of AhLBD proteins was highly conserved, and most AhLBD proteins contained Motif2 and Motif3 near the N-terminus.

To further understand the structural diversity of *AhLBD* genes, the composition of exons/introns in the full-length cDNA of *AhLBD* genes and their corresponding genomic DNA sequences were analyzed (see Figure 4C). The number of introns in *AhLBD* genes ranged from 0 to 7, and genes clustered together usually had similar numbers of introns or exon lengths. A total of 12 *AhLBD* genes without introns were detected, of which 4 were from Class I and 8 were from Class II. Except for *AhLBD1* with 4 introns, *AhLBD62* with 5 introns, and *AhLBD38* with 7 introns, the other genes had only 1–2 introns.

### 2.5. Conserved Sequence Alignment of AhLBDs

Multiple sequence alignment for 73 AhLBD proteins was performed using ClustalW. The analysis showed that all LBD proteins had a LOB region of approximately 100 amino acids at the N-terminus, which was highly conserved (see Figure 5A). All proteins contained a CX2CX6CX3C domain. A total of 26 *AhLBDs* contained a leucine zipper-like domain (LX6LX3LX6L) in Class I and Class II (see Figure 5B), but this domain was not present in Class III, indicating that Class III differs from Class I and Class II in biological function.

### 2.6. Analysis of Cis-Acting Elements in the Promoter Regions of AhLBDs

The cis-acting elements of genes provide strong evidence to support stress responses or tissue-specific expression behaviors in different environments [47]. This study identified all 73 cis-regulatory elements in the *AhLBDs’* promoter region (see Figure 6). The types of cis-regulatory elements mainly include three types, including stress-related, hormone response, and growth and development. The results showed that the distribution and type of cis-acting elements in different subgroups of *AhLBD*s were highly consistent with their gene structure and evolutionary analysis (see Figure 4 and Figure 5). Moreover, *AhLBD* members in the same subgroup were more similar.

In this study, a total of 11 types of cis-acting elements were analyzed. Abscisic acid (ABA) and gibberellin belong to phytohormones response elements. It was found that 60 genes contained 1 to 13 abscisic acid response elements, and 36 genes contained 1 to 3 gibberellin elements (see Figure 6B). All genes consisted of light elements, of which 15 *AhLBD*s contained more than 20 light elements.

Meristem expression elements were detected in 25 *AhLBDs*, and zein metabolism elements were detected in 20 *AhLBDs*. Elements related to abiotic stress response include anaerobic inducible elements, defense stress elements, drought inducible elements, low temperature elements, MeJA response elements, and salicylic acid cis-elements. Among them, 58 *AhLBDs* contained anaerobic inducible elements, 28 *AhLBDs* contained defense stress elements, 19 *AhLBDs* contained drought-induced functional elements, 18 *AhLBDs* contained low temperature elements, 44 *AhLBDs* contained MeJA response elements, and 43 *AhLBDs* contained salicylic acid elements (see Figure 6B). The above results indicated that the peanut plant’s *AhLBD* gene family played a very important role in response to growth and development, abiotic stress response, and plant hormone response. In addition, *AhLBD* genes may have different functions due to the different cis-elements in its promoter region.

### 2.7. Evolutionary Analysis of LBDs between A. hypogaea and Other Species

In order to further explore the evolutionary relationship of the *LBD* gene family with *A. hypogaea* and other species, collinearity analyses of *LBDs* between *A. hypogaea* and *LBD* genes of other species was conducted (see Figure 7), including two dicotyledons (Arabidopsis and soybean) and two monocotyledons (rice and maize). As shown in Figure 7, there were 174 and 47 collinear genes between peanut and soybean and Arabidopsis, respectively as well as 18 and 15 collinear genes between peanut and rice and maize, respectively. These results showed that the evolutionary relationship of *AhLBD* genes with dicotyledonous plants was closer than that with monocotyledonous plants, especially that between peanut and soybean plants.

### 2.8. GO Enrichment Analysis of AhLBDs

In order to predict the biological functions of *AhLBDs*, we performed Gene Ontology (GO) annotation analysis for all 73 AhLBD proteins, revealing that they may participate in a range of cellular components and biological processes (see Figure 8).

The results showed that all AhLBD proteins were assigned a total of 98 GO terms, most of them belong to biological processes, and many proteins were enriched in this category. In the biological processes category, the most highly enriched types were related to lateral root formation, post-embryonic plant organ morphogenesis, lateral root morphogenesis, lateral root development, post-embryonic root development, anatomical structure morphogenesis, and plant organ formation. There were 22 AhLBD proteins that can participate in anatomical structure morphogenesis, and 13 identical AhLBD proteins that can participate in other biological processes mentioned above. Under the cellular component category, the most highly enriched categories were related to the composition of nucleus, of which 19 AhLBD proteins were involved in this component.

In addition, the analysis of the top 20 GO terms were shown in Figure 9, the results showed that the strongest enrichment and the highest enrichment factor (409.5) were observed in proximal/distal pattern formation, followed by petal development and corolla development, of which the enrichment factor for both was 204.8. In addition, the largest number of genes in GO terms was associated with multicellular organism development.

### 2.9. Interaction Network Analysis of AhLBDs in Peanut Plants

To further determine which proteins may interact with AhLBD proteins, we downloaded the Arabidopsis protein interaction network file from the STRING database and used it as the basis for predicting AhLBD-interacting proteins (see Figure 10). The AhLBD protein with other proteins interaction network consisted of 152 nodes, each of which communicated with other nodes. The AhLBD6, AhLBD18, AhLBD41, and AhLBD54 proteins interacted directly with 15 auxin response factors (Ah03g299400.1, Ah05g050200.1, Ah05g315400.1, Ah06g259500.1, Ah06g276200.1, Ah11g439500.1, Ah12g192600.1, Ah13g328900.1, Ah14g207500.1, Ah14g213300.1, Ah14g213400.1, Ah15g050200.1, Ah15g477400.1, Ah16g349300.1, and Ah18g220300.1), and the annotation information of auxin response factors were detailed in Appendix A. The AhLBD7, AhLBD19, AhLBD42, and AhLBD53 proteins each interacted directly with 13 auxin response factors (Ah03g299400.1, Ah05g050200.1, Ah05g315400.1, Ah06g259500.1, Ah11g439500.1, Ah12g192600.1, Ah13g328900.1, Ah14g207500.1, Ah14g213300.1, Ah14g213400.1, Ah15g050200.1, Ah15g477400.1, and Ah18g220300.1). Alongside this, the AhLBD9, AhLBD14, AhLBD15, AhLBD44, and AhLBD50 proteins interacted directly with 8 auxin response factors, respectively (Ah03g299400.1, Ah05g050200.1, Ah11g439500.1, Ah12g192600.1, Ah13g328900.1, Ah14g213400.1, Ah14g213300.1, and Ah15g050200.1). However, there was no direct interaction between AhLBD proteins. Protein interaction network analysis only predicted the potential interacting proteins of AhLBD family genes, and protein interaction experiments were still needed for verification.

### 2.10. Expression Profiles of AhLBDs in Different Developmental Stages and Abiotic Stress

To further determine which *AhLBD* genes potentially function in different tissues, the expression profile of 73 *AhLBD* members were investigated by using transcriptome data with 21 tissues and organ samples at different developmental stages of *A. hypogaea* (*Tifrunner*) (accession: PRJNA291488). The heat map shows the expression of *AhLBD* genes in different tissues and organs (see Figure 11). The results showed that this family of genes has constitutive, tissue-specific, and low expression patterns, indicating that there is significant diversity in the expression profile of the *AhLBD* gene family. From the transcriptome heatmap, we eliminated 41 *AhLBD* genes in the three groups whose FPKM values were almost zero in any tissue or organ.

The traits related to root development included root and nodule roots. The expression levels of *AhLBD67*, *AhLBD52*, and *AhLBD17* were relatively high in the root (see Figure 11). The traits related to stem development included the vegetative shoot tip. The expression level of *AhLBD67* in the stem was much higher than other genes. The traits related to leaf development included MainLeaf, lateral leaf, and leaf. *AhLBD71* had the highest expression level in leaves, followed by *AhLBD67* and *AhLBD33*, which also had higher expression levels in leaves. *AhLBD71* had the highest expression levels in flowers and pistil, while *AhLBD51* and *AhLBD67* had higher expression levels in stamen. The expression level of *AhLBD51* was much higher in pericarp patee 5 and pericarp pattee 6 than in other genes. Among seed pattee 5, seed pattee 6, seed pattee 7, seed pattee 8, and seed pattee 10, *AhLBD67* had the highest expression level. From the expression analysis of the above 21 organs, *AhLBD67* had the highest expression level in 11 tissues and organs, including root, nodule roots, vegetative shoot tip, aerial peg, seed pattee 5, seed pattee 6, seed pattee 7, seed pattee 8, and seed pattee 10, indicating that *AhLBD67* plays an important role in root, stem, and seed development. Alongside this, *AhLBD71* had the highest expression levels in flowers, pistil, MainLeaf, lateral leaf, and leaf, indicating that *AhLBD71* might play an important role in reproductive organs and leaves.

The expression level of *AhLBD* genes under cold (4 °C), heat (37 °C), salt (2% NaCl), and drought (20% PEG6000) stress for 24 h treatment were also detected (see Figure 12). The results showed that some *AhLBD* members were related to the regulation of abiotic stress, and different genes showed different expression trends. It was mainly concentrated in Class I and Class II for the four abiotic stresses. In addition, the expression of 31 out of all 73 *AhLBD*s were not detected under any abiotic stress.

Under salt stress, the expression levels of *AhLBD67* in roots and leaves were similar and much higher than other genes (see Figure 12). Under drought stress, the expression levels of *AhLBD67* in roots and leaves were similar and higher than other genes. *AhLBD71*, *AhLBD33*, *AhLBD52,* and *AhLBD17* were expressed much more in roots than in leaves, while *AhLBD56* was expressed at 0 in roots but higher in leaves. After high temperature treatment, the expression of *AhLBD67* was the highest in roots and leaves, and the expression of *AhLBD35* and *AhLBD72* was relatively high in roots; *AhLBD52*, *AhLBD17*, *AhLBD56*, *AhLBD71*, *AhLBD24*, *AhLBD33*, and *AhLBD59* gene expression was relatively high in leaves. After cold treatment, *AhLBD71* had the highest expression level in leaves, followed by *AhLBD17*, *AhLBD52*, *AhLBD33*, and *AhLBD67*. In roots, *AhLBD16*, *AhLBD51*, and *AhLBD67* had higher expression levels. In conclusion, *AhLBD67*, *AhLBD71*, and *AhLBD16* may play an important role in abiotic stress.

### 2.11. qRT-PCR Identification of 12 Selected AhLBDs in Different Tissues

In order to prove the reliability of the RNA sequencing data, 12 genes with significantly different expression levels during the development period were further selected for qRT-PCR verification (see Figure 13). The expression analysis results of the tested *AhLBDs* in different tissues were basically consistent with the RNA sequencing data. The expression level of *AhLBD12/17/52* genes in roots were significantly higher than that in other tissues. And *AhLBD24/59/67/72* genes had significantly higher expression in stems. Alongside this, the expression level of *AhLBD33/35/71* were significantly higher in flowers, and *AhLBD51* gene had the higher expression in leafs and seed1, while the expression level of *AhLBD16* was significantly higher in seed1.

### 2.12. qRT-PCR Analysis of 12 AhLBDs under Salt and Drought Stresses in Peanut Plants

To further confirm the expression levels of *AhLBDs* under salt and drought stress, the peanut variety “*Qinghua 6*” was treated with NaCl and PEG, respectively, and qRT-PCR analysis of 12 *AhLBDs* was performed (see Figure 14).

Under salt stress, the expression levels of *AhLBD12/16/33/52/67/71* genes increased compared with 0 h, while *AhLBD17/24/35/51/59/72* decreased compared with 0 h, respectively (see Figure 14A). Among the genes with increased expression, the expression level of *AhLBD12/52/71* was the highest at 12 h, respectively, while that of *AhLBD16/33/67* were the highest at 24 h. The expression level of *AhLBD12/16/33/52/67* increased significantly at 12 h compared with 0 h, and that of *AhLBD16/33/67* increased significantly at 24 h. Among the genes whose expression decreased, *AhLBD24/51/59/72* genes had the lowest expression level at 3 h, while *AhLBD17/35* genes had the lowest expression level at 6 h.

Under drought stress, the expression levels of seven genes (*AhLBD24/33/35/59/67/71/72*) increased, while the expression levels of five genes (*AhLBD12/16/17/51/52*) decreased compared with 0 h (see Figure 14B). Among the genes with increased expression, *AhLBD59/72* had the highest expression level at 3 h, *AhLBD24/33/67/71* had the highest expression level at 12 h, and *AhLBD35* had the highest expression level at 24 h. Their expression levels were all significantly increased compared with 0 h. Among the genes with reduced expression, *AhLBD51* had the lowest expression at 3 h, *AhLBD12/17* had the lowest expression at 6 h, and *AhLBD16/52* had the lowest expression at 24 h. In addition, the expression of *AhLBD17* at 6 h was significantly lower than that at 0 h, while the expression level of *AhLBD16/52* had significantly reduced at 24 h.

### 2.13. Subcellular Localization Analysis of Four AhLBDs

As we know, the localization of a protein in a cell is closely related to its function. To explore the function of *AhLBD24*, *AhLBD33*, *AhLBD67,* and *AhLBD71*, the full-length coding sequences of the above four genes were cloned into the pRI101-35S-GFP vector and transiently transformed in *Nicotiana benthamiana* leaves. The subcellular localization results showed that the fusion proteins AhLBD24-GFP, AhLBD33-GFP, AhLBD67-GFP, and AhLBD7-GFP were all detected in the nucleus (see Figure 15). This suggested that the four genes function in the nucleus, which is consistent with subcellular localization prediction results (see Appendix A).

### 2.14. Overexpression of AhLBD33 and AhLBD67 in Arabidopsis thaliana and Salt Stress Test

To further verify the functions of *AhLBD33* and *AhLBD67* genes, three overexpressed *Arabidopsis* T3 generation homozygous lines were selected for salt stress experiments (see Figure 16). The results showed that under 200 mM NaCl stress treatment, the main root lengths of *AhHDZ33* overexpression lines OE1 and OE6 were 3.48 cm and 3.42 cm, respectively, which were significantly longer than WT (2.74 cm). Moreover, the main root length of the OE2 line was 3.55 cm, which was significantly longer than the WT.

In addition, under salt stress treatment, the main root length of *AhLBD67* overexpression line OE3 was 3.11cm, which was significantly longer than WT (2.57 cm). Moreover, the main root lengths of OE2 and OE6 lines were 3.34 cm and 3.35 cm, respectively, which were significantly longer than that of the WT by even larger margins.

The above results show that the salt tolerance of Arabidopsis overexpressing *AhLBD33* and *AhLBD67* was significantly enhanced, indicating that *AhLBD33* and *AhLBD67* play an important role in the salt tolerance of peanut plants.

## 3. Discussion

### 3.1. Comparison of Phylogenetic Relationships between the Peanut Plant LBD Family Genes and Other Species

The LBD protein family is a plant-specific transcription factor that plays an important role in plant growth and development and stress. However, no systematic study of LBD genes has been conducted in peanut plants. In this study, a total of 73 members were identified from the whole genome of the tetraploid peanut cultivar (*Tifrunner*). Compared with the number of LBD members reported in *Arabidopsis thaliana* (34) [35], soybean plants (92) [39], corn (44) [41], rice (31) [42], barley (34) [43], *Pinus Massoniana* (47) [44], *Mosobamboo* (55) [45], and sweet potato plants (53) [46], peanut plants have the largest number of family members. Combining the analysis of the number of LBD family genes and genome size of each species, we found that there was no correlation between the number of LBD family members and genome size in different species. Moreover, phylogenetic tree analysis showed that the *AhLBD* gene family was divided into three groups, of which the first group had 53 *LBD* members, accounting for 72.6%, which was different from other species such as *Arabidopsis* (see Figure 4). In addition, studies have shown that gene family expansion mainly depends on tandem, fragment and whole genome replication forms [48]. Combined with the analysis of *LBD* gene family members in these species, it is speculated that this phenomenon may be caused by the different frequency of fragment duplication and tandem duplication among different species during the evolution of the LBD gene family. Furthermore, the collinearity results of peanut plants with monocotyledonous plants (rice and maize) and dicotyledonous plants (*Arabidopsisa* and soybean) showed that peanut had significantly more collinearity with *Arabidopsis* and soybean than with rice and corn, indicating that the genetic relationship between peanut and dicotyledonous plants was closer.

Studies had shown that the number of exons of *LBD* homologous genes in maize and rice was 1–5 and 2–4, respectively [41,42], and the analysis of the gene structure of *AhLBD*s in this study showed that the number of exons of most genes was also 1–3 (except *AhLBD1*, *AhLBD38*, and *AhLBD62*), indicating that the *LBD* gene family structure among these species was relatively simple and may be relatively conserved during evolution.

### 3.2. The Potential Biological Functions of AhLBDs in Peanut Plants

Promoter *cis*-acting elements were closely related to gene expression regulation [49]. This study found that there are many cis-acting elements in the promoter region of the peanut plant’s LBDs gene, which could be divided into three categories: growth and development related elements, light regulation, and stress response elements. Among them, light regulatory elements were detected in all *AhLBD* family genes. Abscisic acid responsive elements and anaerobic induction essential elements were also detected in most genes, suggesting that light and abscisic acid might affect the expression of *AhLBDs*, thereby affecting the growth and development of peanut plants. In addition, it should be noted that *AhLBD4*, *AhLBD29,* and *AhLBD53* contained more light-responsive elements than other LBD family members in peanut plants, and it was speculated that these genes might respond more strongly to light than other genes. Both *AhLBD35* and *AhLBD72* in Class I had three drought-inducibility response elements (see Figure 6), and fluorescence quantitative PCR results showed that *AhLBD35* and *AhLBD72* responded to drought stress, which might be related to the fact that they contain more drought-inducibility cis-acting elements.

In addition, more than half of the top 20 GO items enriched in *AhLBDs* were related to the development and formation of plant organs (see Figure 9). Moreover, the results of potential interactions between AhLBD and other proteins from the peanut plant showed that *AhLBD6/7/14/15/18/19/41/42/44/50/53/54* interacted directly with auxin response factors, suggesting that these genes might be involved in the growth and development of people. So far, many studies have shown that *LBD* genes also play an important role in abiotic stress response [48,49,50]. The RNA-Seq data and qRT-PCR results in our study both showed that many *AhLBD* genes responded to salt and drought stress (see Figure 12 and Figure 14). These results indicate that *AhLBD* genes are widely involved in various abiotic stress responses. In addition, it is well known that roots are the main organs responding to drought and salt stress [51].

In this study, the expression of *AhLBD16, AhLBD33,* and *AhLBD67* was increased by more than 3 times in the roots after 24 h of salt treatment. These genes may be candidate genes for positive regulation of peanut plant salt stress. Compared with 0 h salt treatment, the expression of *AhLBD24* and *AhLBD51* were significantly down-regulated during 3–6 h salt treatment, and the expression of *AhLBD72* was significantly down-regulated during 3–24 h salt treatment. This indicates that these genes may be negative regulatory genes in response to salt stress in peanut plants. This study further confirmed through molecular genetic experiments that heterologous overexpression of *AhLBD33* and *AhLBD67* in *Arabidopsis* could significantly enhance tolerance to salt stress.

In addition, at 12 h and 24 h, the expression levels of *AhLBD24* and *AhLBD71* were upregulated 6 times compared with that of before treatment; the expression levels of *AhLBD33* and *AhLBD67* were upregulated 3 and 4 times, respectively; and the expression level of AhLBD35 was also upregulated 3 times. This indicates that these genes are candidate genes for the positive regulation of peanut plant drought stress.

Furthermore, subcellular localization results showed that *AhLBD24*, *AhLBD33*, *AhLBD67*, and *AhLBD71* were nuclear-localized genes, further proving that these genes are very likely to play the role of transcription factors in abiotic stress. Previous studies have shown that *LBD* genes of multiple species are significantly upregulated under different abiotic stresses. For example, *Ta-6B-LBD 81*, *Ta-4B-LBD51,* and *Ta-U-LBD90* were found to be upregulated under salt stress, while *Ta-2A-LBD13*, *Ta-2B-LBD15,* and *Ta-2D-LBD18* were found to be upregulated under drought stress [52]. Moreover, potato genes *StLBD2-6* and *SLBD3-5* were found to be significantly upregulated under drought stress [40]. In conclusion, *AhLBD* genes showed different responses to salt stress and drought stress. Although the molecular mechanism of *AhLBD* involved in abiotic stress response are still unclear, this study has provided a key candidate gene for further resistance research in peanut plants.

## 4. Materials and Methods

### 4.1. Identification and Physicochemical Properties of the AhLBD Gene Family

The *A. hypogaea Tifrunner* genome v2.0 was used in this study. The protein sequences of *AtLBD* family genes were obtained from the TAIR database (https://www.arabidopsis.org/, accessed on 15 March 2024). The genomic data and GFF3 file of *A. hypogaea* were obtained from the PeanutBase website (https://dev.peanutbase.org/, accessed on 15 March 2024). All AtLBD protein sequences were used as the queries to perform a BLASTP search against the local protein database of *A. hypogaea* with an e-value 1e-10and identity of 50% as the threshold. Furthermore, the *LBD* domain (PF03195) obtained from the PFAM database (http://pfam.xfam.org/, accessed on 15 March 2024) was used as the query for the hidden Markov model (HMM) search using the HMMER 3.0 program (http://hmmer.org/, accessed on 15 March 2024) [53]. The AhLBD protein sequences identified by blast and HMM searching were integrated and parsed by manual editing to remove redundancy, and the remaining genes were considered as the putative *AhLBDs*. The ExPASy ProtParam tool (http://web.expasy.org/protparam/, accessed on 15 March 2024) was used to predict protein physicochemical parameters [54]. Alongside this, the subcellular location of *AhLBDs* was predicted by WOLF–PSORT (https://wolfpsort.hgc.jp/, accessed on 15 March 2024) [55] and CELLO ver.2.5 (http://cello.life.nctu.edu.tw/, accessed on 15 March 2024) [56].

### 4.2. Phylogenetic Analysis of AhLBD Proteins

ClustalW software in MEGA11.0 was employed for a multiple sequence alignment of LBD protein sequences between the peanut plant and Arabidopsis. The phylogenetic tree was constructed with the Maximum Likelihood method with 1000 bootstrap replications using MEGA11.0 [57]. The trees were further modified using EvolView (https://www.evolgenius.info/evolview, accessed on 15 March 2024) [58].

### 4.3. Analysis of Chromosome Location and Synteny Analysis of AhLBDs

The length of each chromosome and the position information of *AhLBDs* on chromosomes were extracted from the GFF3 file of *A. hypogaea,* and then mapped by Mapchart [59]. The synteny relationship of *LBDs* between peanut plants and Arabidopsis, soybean plants, rice, and maize were performed by using MCScanX (https://github.com/wyp1125/MCScanX, accessed on 20 April 2024) [60]. Circos and the dual synteny plot tool in TBtools-II (v2.043) were used for visualized mapping of the collinear gene pairs [61].

### 4.4. Gene Structure, Conserved Motif, and Promoter Cis-Element Analysis

The exon–intron distributions for all *AhLBD* genes were obtained using GFF3 annotation files from the *A. hypogaea* genome. Conserved amino acid sequences of LBD proteins were analyzed using the online MEME tool (http://meme-suite.org/, accessed on 15 March 2024) [62]. MEME analysis parameters included a minimum width ≥6, a maximum width of 50, and a motif number of 10, and all other parameters were set to default values. The 2000 bp upstream sequences of the start codon (ATG) of *AhLBDs* were chosen to identify the *cis*-acting elements in the putative promoter regions, which were identified with the online program PlantCARE (http://bioinformatics.psb.ugent.be/webtools/plantcare/html/, accessed on 15 March 2024) [63]. The phylogenetic tree along with *cis*-acting elements was visualized by using TBtools-II (v2.043)software [61].

### 4.5. Protein–Protein Interaction Network and Gene Ontology (GO) Analysis of AhLBDs

We downloaded the Arabidopsis protein database and the Arabidopsis protein interaction relationship file from the STRING database (https://stringdb.org/, accessed on 15 March 2024). We then used TBtools-II software to compare and screen the AhLBD family peanut plant interaction protein network file through the Arabidopsis protein interaction relationship file [61]. Finally, we used Cytoscape software (v3.10.2) to visualize the interaction protein network [64]. The file of go-basic.obo was downloaded from the website (http://purl.obolibrary.org/obo/go/go-basic.obo/, accessed on 15 March 2024). GO annotation was analyzed using TBtools-II(v2.043) [61] and visualized with the online software tool (https://www.bioinformatics.com.cn/, accessed on 15 March 2024) [65].

### 4.6. RNA-Seq Based Expression Profiling of AhLBDs

The RNA-seq data of 21 distinct tissue types and developmental stages (vegetative shoot tip, reproductive shoot tip, root, nodule roots, flowers, pisti, stamen, aerial peg, subterranean peg, MainLeaf, lateral leaf, leaf, pod pattee 3, pericarp pattee 5, pericarp pattee 6, seed pattee 5, seed pattee 6, seed pattee 7, seed pattee 8, seed pattee 10, and gyn tip stalk) (accession: PRJNA291488), along with cold (4 °C), heat (37 °C), salt (2%Nacl) and drought (20% PEG6000) treatments (accession: PRJNA553073) were downloaded from NCBI database (https://www.ncbi.nlm.nih.gov/, accessed on 15 March 2024). All gene expression levels were normalized using log_2_(FPKM+1). The heat maps were generated through TBtools-II (v2.043)software [61].

### 4.7. Plant Material, Abiotic Stress Treatments and qRT-PCR Analysis

*A. hypogea* cultivar “*QingHua 6*” was grown in the experimental field at the Institute of Cotton Research of Shanxi Agricultural University (Yuncheng, China). Peanut plant tissue samples at different developmental stages were collected, including root, stem, young leaves at the seedling stage, and flower at the flowering stage, and seeds developed 40 and 70 days after flowering were collected.

*QingHua 6* was also grown in a greenhouse with 24–26 °C (16 h light/8 h dark). Six leaf seedlings of peanut plants were grown in 1/2 Hoagland solution and were treated with salt (2% NaCl) and drought (30% PEG6000). PEG6000 is an ideal osmotic regulator for simulating drought stress [66]. The control was collected without any treatment at the same stage. The roots were collected at different time points (3, 6, 12, and 24 h after treatment), frozen in liquid nitrogen immediately, and then stored at −80 °C until RNA extraction. Three replicate samples for each treatment were collected.

To further evaluate the expression patterns of the selected *AhLBD* genes in different peanut plant tissues and various abiotic stresses, RNA was extracted using an RNAprep Pure Plant Kit (Tiangen, Beijing, China) and first-strand cDNA was synthesized using a Prime Script^®^ RT reagent kit (Takara, Dalian, China). qRT-PCR was carried out with three biological replicates using a TBGreen Premix Ex Taq Kit (Takara, Dalian, China) on an Applied Biosystems QuantStudio 5 Real-time PCR System (ABI, Thermo Fisher Scientific, USA). *AhACT11* (*Ah10g066400.1*) was used as an endogenous control for qRT-PCR. Relative gene expression levels were calculated using the 2^−ΔΔCT^ method [67]. The primers for qRT-PCR are listed in Appendix A.

### 4.8. Subcellular Localization Assay

For subcellular localization, the full-length coding sequences of genes *AhLBD24*, *AhLBD33*, *AhLBD67*, and *AhLBD71* were cloned and ligated into the pRI101-35S-GFP vector (primers listed in Appendix A), while the empty pRI101-35S-GFP vector was used as a control. The fusion vector was then transformed into *Agrobacterium tumefaciens* strain GV3101. *Agrobacterium*-mediated transient transformation of *Nicotiana benthamiana* leaves was applied according to Sparkes et al. (2006) [68]. The injected tobacco leaves were cultured for 36–48 h. The GFP fluorescence in the leaves was observed using confocal microscopy (Leica, SP8, Germany), GFP was excited at 488 nm and captured at 500–530 nm.

### 4.9. Transformation of Arabidopsis thaliana Overexpressing AhLBD33 and AhLBD67 and Salt Stress

The *AhLBD33* and *AhLBD67* subcellular localization vectors constructed above were used as overexpression vectors and transformed into *Agrobacterium tumefaciens* strain GV3101. *Arabidopsis thaliana* (*Columbia ecotype*) was inoculated using the floral dipping method [69]. After inoculation, the plants were dark-treated for 24 h and then transferred to an incubator for growth at 20–22 °C with 16 h of light/8 h of darkness. They were inoculated again after 7 days until the T_0_ generation seeds were harvested. Positive strains of T_0_-T_3_ generation seeds were screened on 1/2 MS solid culture plates containing Kan^+^ (50 μg/mL). Three T3 generation overexpression transgenic lines and wild-type materials were selected. When they were grown on the non-resistant 1/2 MS medium for 5 days, the uniformly growing seedlings were transferred to a new square plate containing 1/2 MS medium containing 200 mmol/L NaCl. Each square plate contained three homozygous overexpression lines and WT, with three replicates. A total of 10 seedlings were transferred to each replicate for stress testing. After 7 days of vertical cultivation in a light incubator, the taproot was photographed and the root length was measured using ImageJ software. (https://imagej.net/ij/download.html, accessed on 20 April 2024).

## 5. Conclusions

In this study, the *LBD* gene family of peanut plants was systematically identified and analyzed. A total of 73 *AhLBDs* were identified and divided into three groups by phylogenetic tree analysis. Gene structure and conserved protein motif analysis supported the evolutionary conservation of *AhLBDs*. The chromosomal locations, gene duplications, promoters, and PPI network of the peanut *LBD* genes were also investigated. Gene expression profiling and qRT-PCR analysis showed that different genes have different responses to salt and drought stresses. There were also differences in expression patterns. *AhLBD24*, *AhLBD33*, *AhLBD67*, and *AhLBD71* were all located in the nucleus, which indicated that these genes play a role as transcription factors in plant growth and abiotic stress. Heterologous overexpression of *AhLBD33* and *AhLBD67* could significantly enhance the salt tolerance of transgenic *Arabidopsis*. These results provide important candidate genes and a theoretical basis for studying the molecular mechanism for abiotic stress in peanut plants.

## Figures and Tables

**Figure 1 ijms-25-10561-f001:**
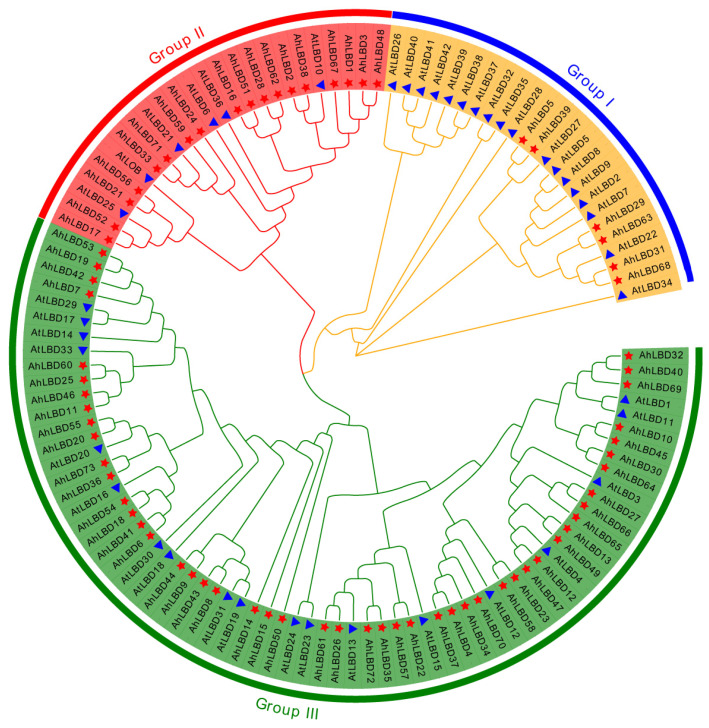
Phylogenetic tree of LBDs between *A. hypogaea* and *A. thaliana*. The Arabidopsis LBD protein sequences were downloaded from the TAIR database. The phylogenetic tree was constructed by the maximum likelihood method based on MEGA11 with 1000 bootstrap replicates performed. Red pentagram and blue triangle represent the *peanut* and *Arabidopsis LBD* genes.

**Figure 2 ijms-25-10561-f002:**
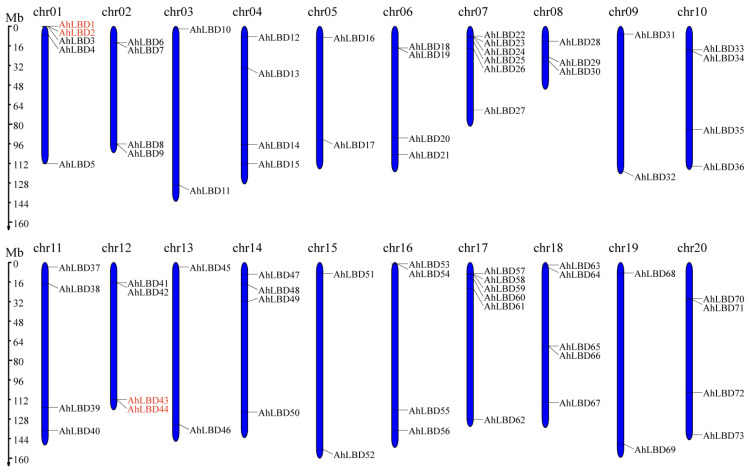
Chromosomal distribution of the *AhLBD* gene family. The scale on the left is in mega-bases. The gene ID on the right side of each chromosome corresponds to the approximate locations of each *AhLBD* gene.

**Figure 3 ijms-25-10561-f003:**
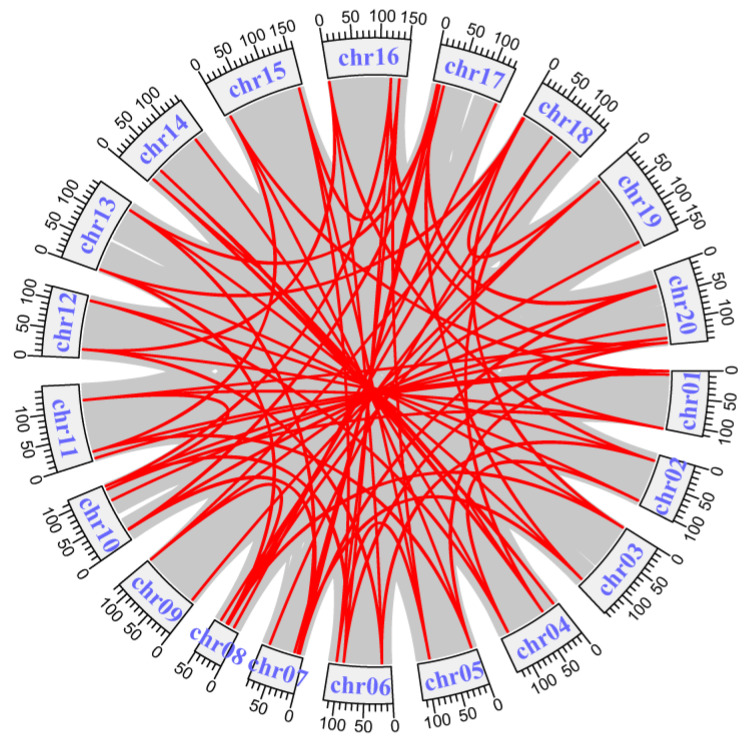
Collinearity analysis of the *AhLBD* gene family in *A. hypogaea*. Red lines represent colinear gene pairs of *AhLBDs*.

**Figure 4 ijms-25-10561-f004:**
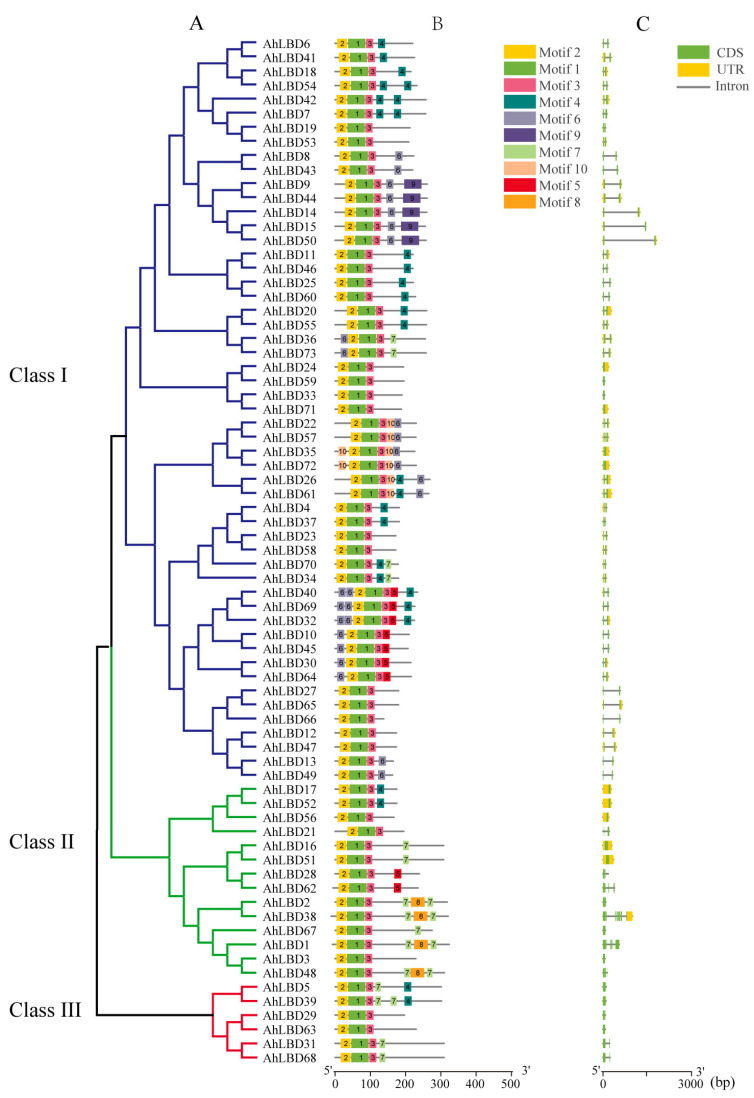
Phylogenetic tree, conservative motif, and gene structure analysis of all *AhLBDs*. (**A**) A maximum likelihood (ML) phylogenetic tree of *A. hypogaea* protein with 1000 bootstrap replicates was constructed based on the full-length sequence in MEGA11. (**B**) Distribution of conservative motifs in AhLBD proteins with colored boxes representing motifs 1–10. (**C**) The genetic structure of the *AhLBD* genes, including CDS (green rectangle), UTR (untranslated region, yellow rectangle) and intron (black line).

**Figure 5 ijms-25-10561-f005:**
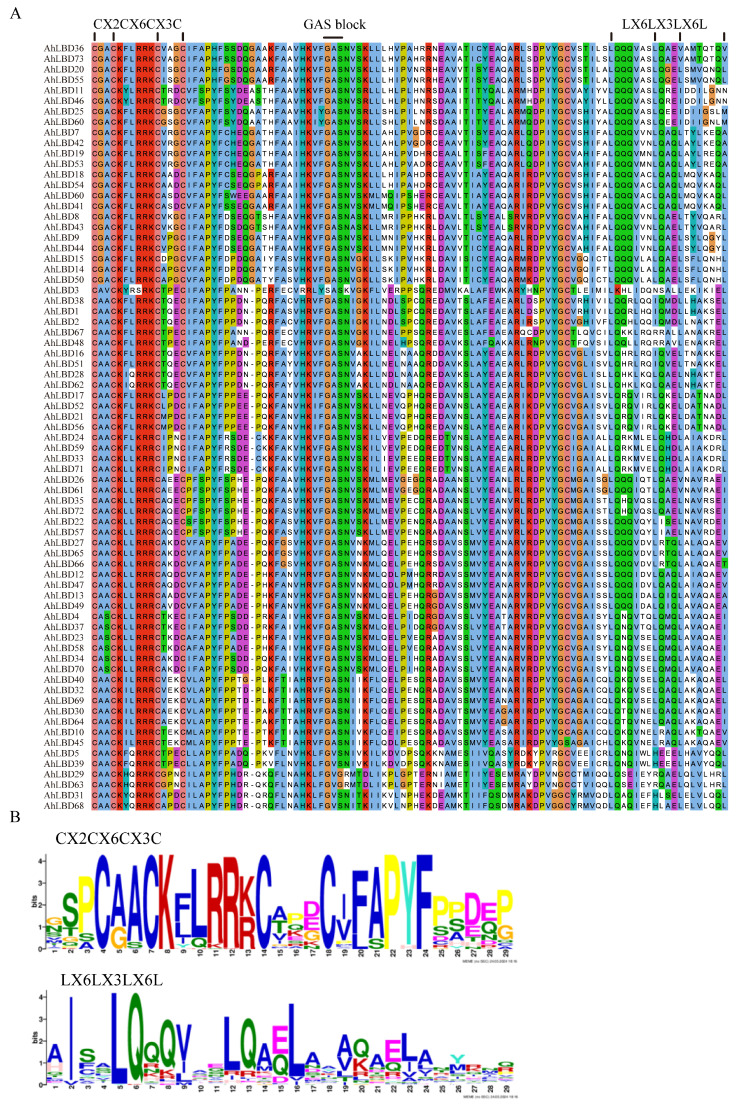
Multiple sequence alignment and conservative domains of AhLBD proteins. (**A**) The zinc finger domain (CX2CX6CX3C) was present in all 73 predicted AhLBD protein sequences, while the leucine zipper-like motif (LX6LX3LX6L) was present in Class I and II AhLBD proteins. (**B**) Alignment of conservative motifs generated by the MEME online website for the two protein domains.

**Figure 6 ijms-25-10561-f006:**
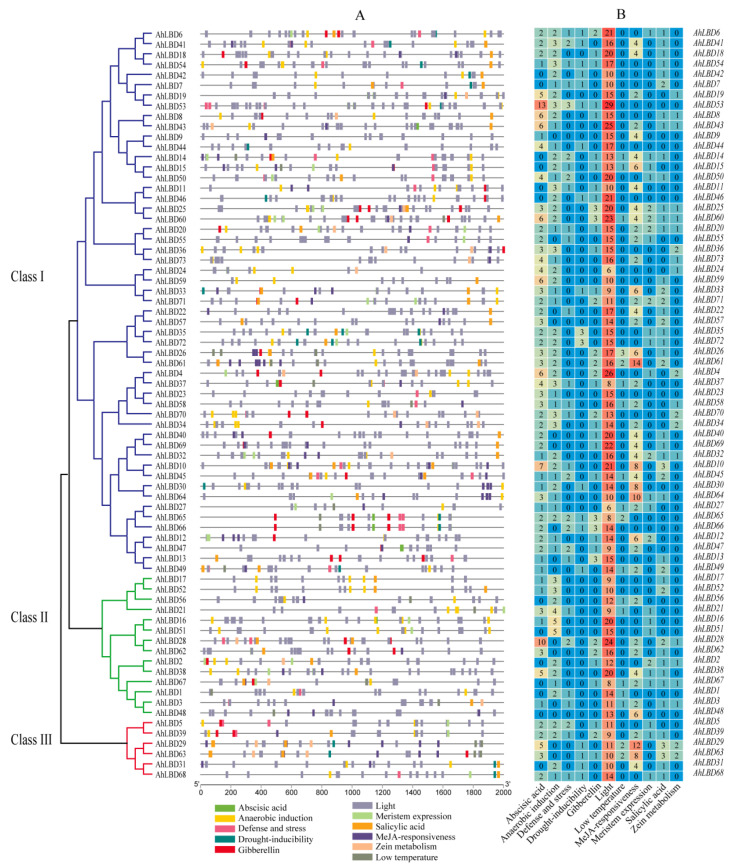
Distribution of *cis*-acting elements of *AhLBD* gene family in the peanut plant. (**A**) Distribution of *cis*-acting elements identified in the 2000 bp upstream promoter region of start codon (ATG) of *AhLBD* genes. (**B**) Heatmap of the number of cis-acting element types for *AhLBDs*.

**Figure 7 ijms-25-10561-f007:**
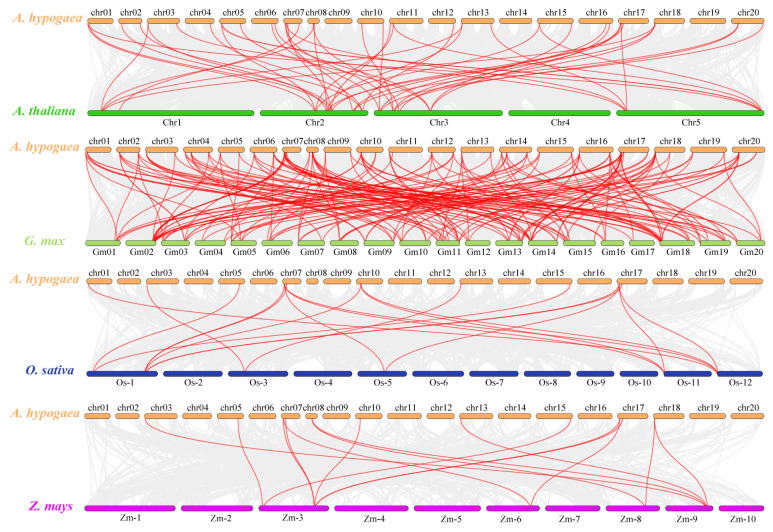
Collinearity analysis of LBDs between *A. hypogaea* and other species. The species are Arabidopsis, soybean, rice, and maize. The red line represents the homologous *LBD* gene pairs of the plant genome, while the gray line represents the collinear block of the plant genome.

**Figure 8 ijms-25-10561-f008:**
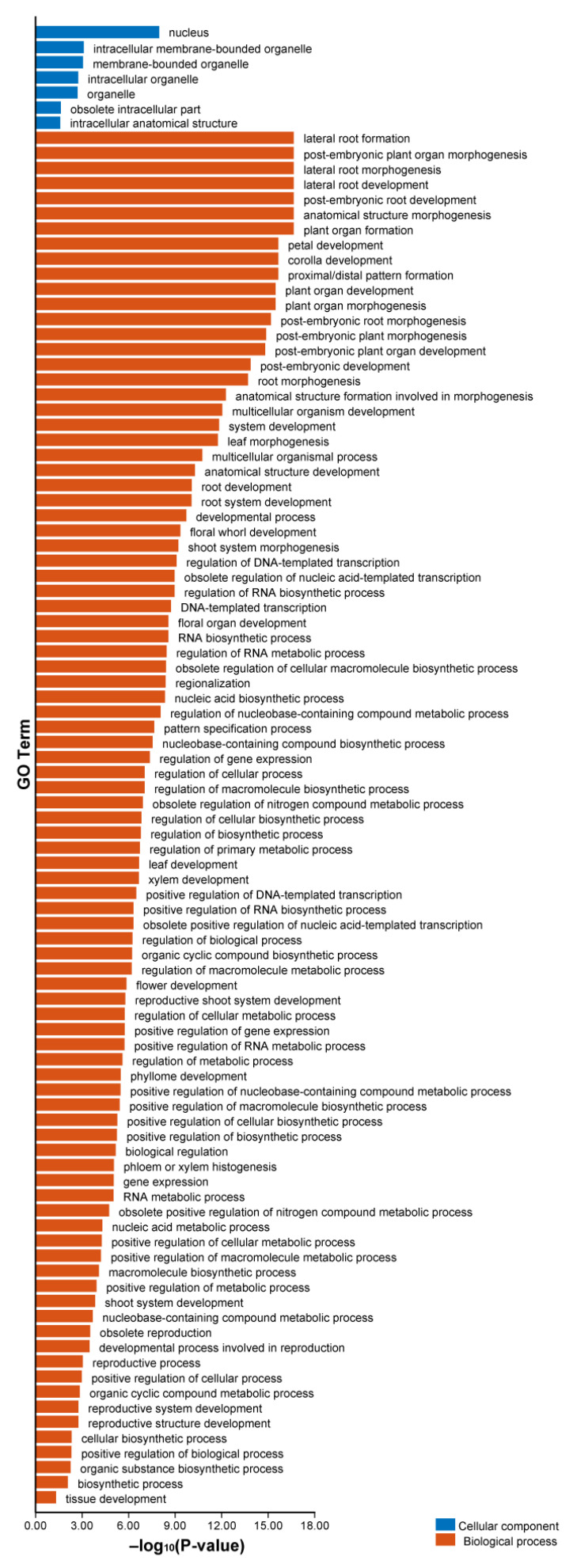
GO annotations for AhLBD proteins. The GO annotation was divided into two categories: biological process and cellular component. The *x*-axis represents GO Term; the *y*-axis represents −log_10_ (*p*-value) of the GO annotation.

**Figure 9 ijms-25-10561-f009:**
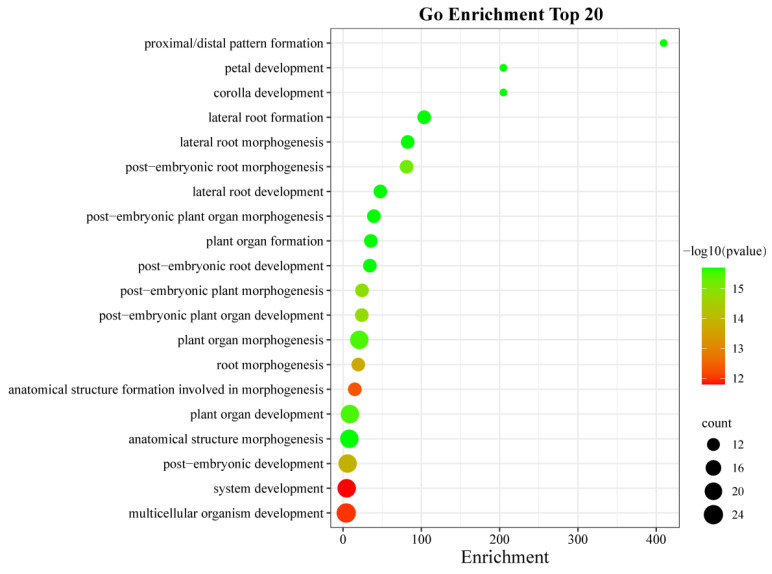
The top 20 enriched GO terms of candidate *AhLBD* target genes. The black circles indicate the number of target genes, and different colors indicate the –log10 (*p*-value), ranging from 12 to 15.

**Figure 10 ijms-25-10561-f010:**
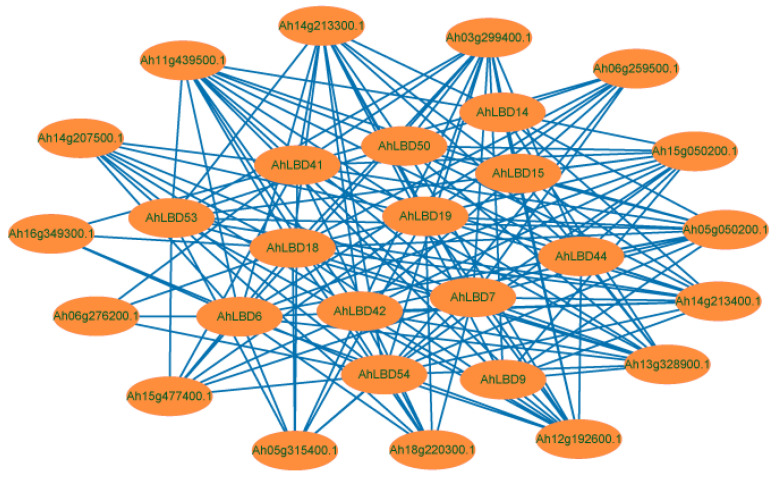
The AhLBDs functional interaction networks of peanut based on *Arabidopsis* orthologs. Proteins serve as network nodes and protein–protein relationships are represented by lines.

**Figure 11 ijms-25-10561-f011:**
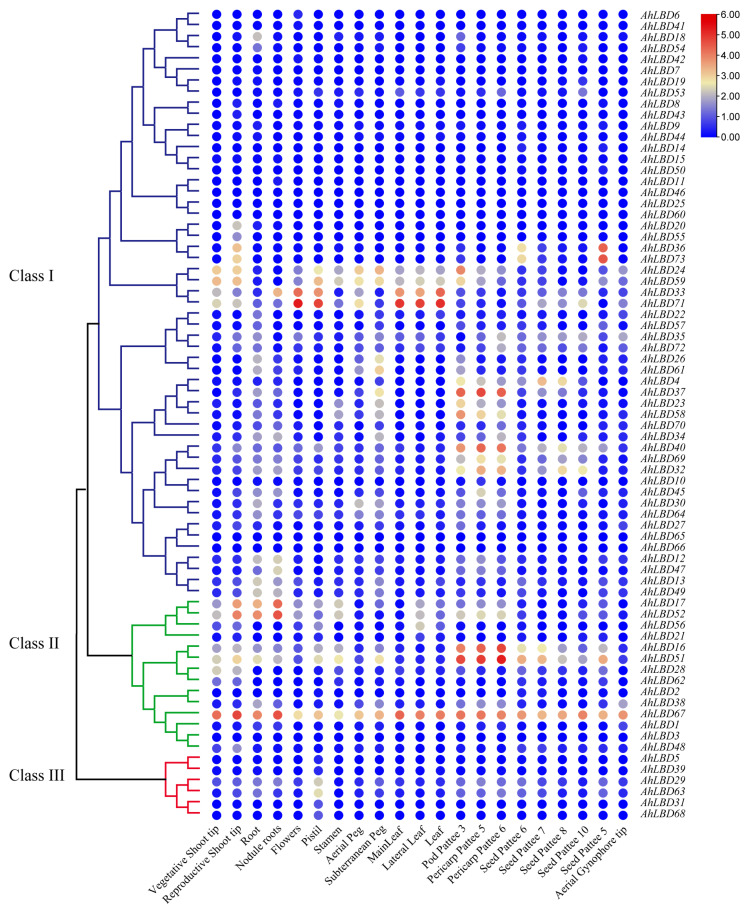
Expression profiles of the *AhLBD* gene family in different tissues. The heat maps are constructed based on transcriptome data (FPKM values). Different colors represent the different expression levels of *AhLBDs*. Gene names are shown on the right. The scale bar on the right shows the colors used to describe the log_2_ (FPKM+1) for each gene. FPKM, fragments per kilobase of transcript per million mapped reads. Pod Pattee 3: Pericarp very watery, embryo very small and not easily removed (Pattee stage 3/4); Pericarp Pattee 5: Pericarp soft, not as watery, inner pericarp without cracks (Pattee stage 5); Pericarp Pattee 6: Inner pericarp tissue beginning to show cracks or cottony (Pattee stage 6/7); Seed Pattee 5: Embryo flat, white or just turning pink at one end (Pattee stage 5); Seed Pattee 6: Torpedo shaped; generally pink at embryonic-axis end of kernels (Pattee stage 6); Seed Pattee 7: Torpedo to round shaped; embryonic axis end of kernel pink; other end white to light pink (Pattee stage 7); Seed Pattee 8: Round, light pink all over (Pattee stage 8); Seed Pattee 10: Large, generally dark pink all over; seed coat beginning to dry out (Pattee stage 10).

**Figure 12 ijms-25-10561-f012:**
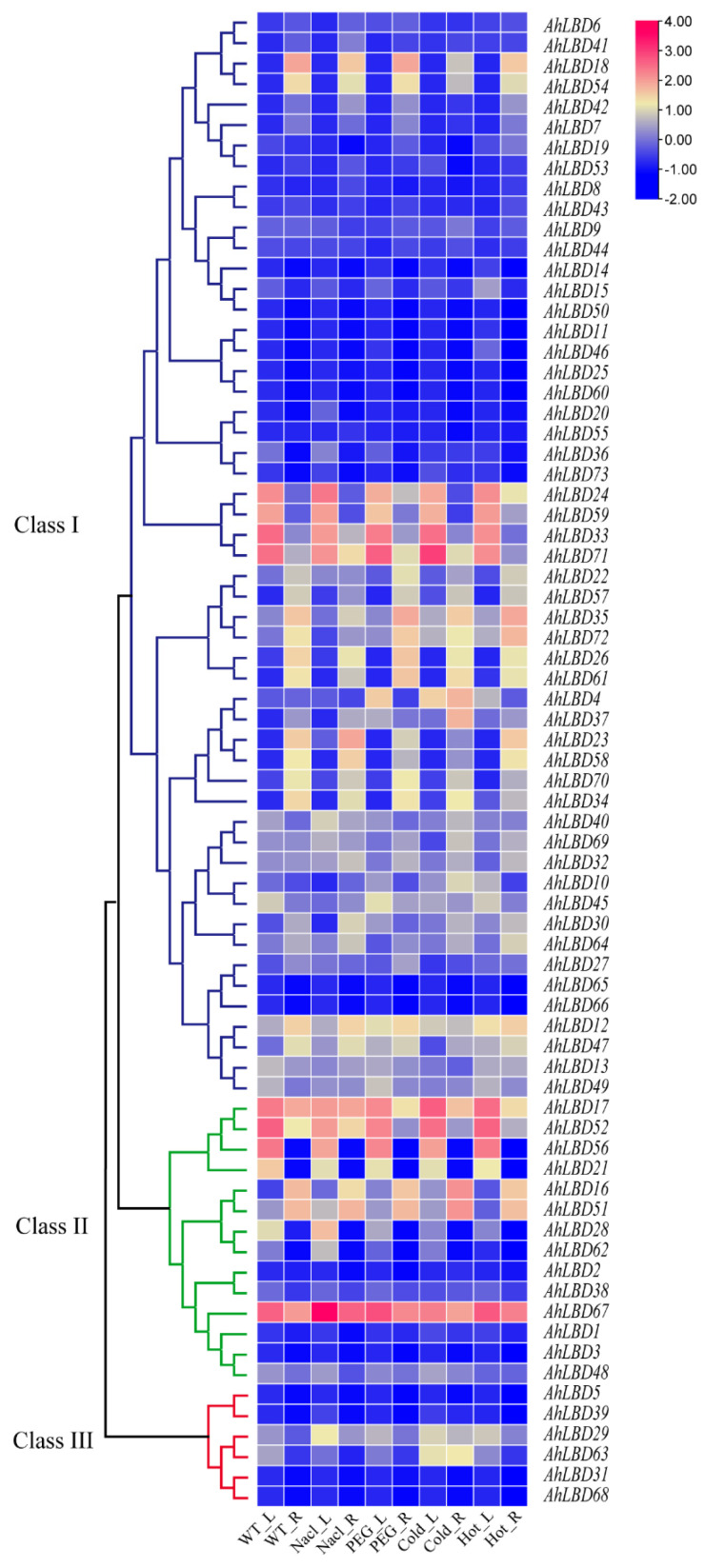
Expression profiles of *AhLBDs* under 2% Nacl, 20%PEG 6000, 4 °C cold, 37 °C hot with leaf and root. The gene names are shown on the right. Class I, II, and III represent three groups according to the phylogenetic tree. Scale bars on the right represent log_2_ (FPKM+1).

**Figure 13 ijms-25-10561-f013:**
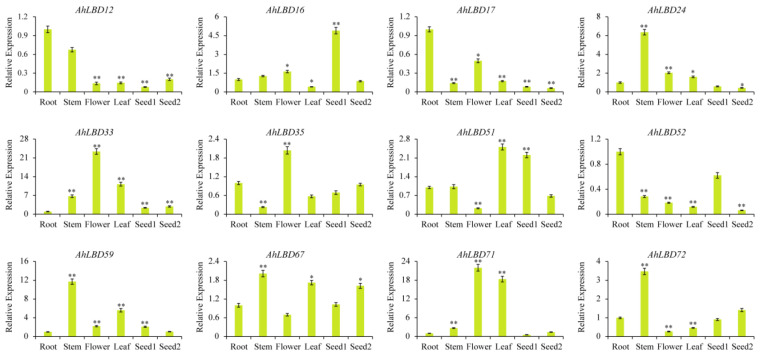
Expression patterns of 12 *AhLBD* genes in different tissues. Error bars indicate standard deviations among three independent biological replications. *AhACT11* was used as the internal control. Seed1: Pattee 5 seed; Seed2: Pattee 10 seed. *t*-test was used to analyze three biological replicates of each sample. *: *p* < 0.05, **: *p* < 0.01.

**Figure 14 ijms-25-10561-f014:**
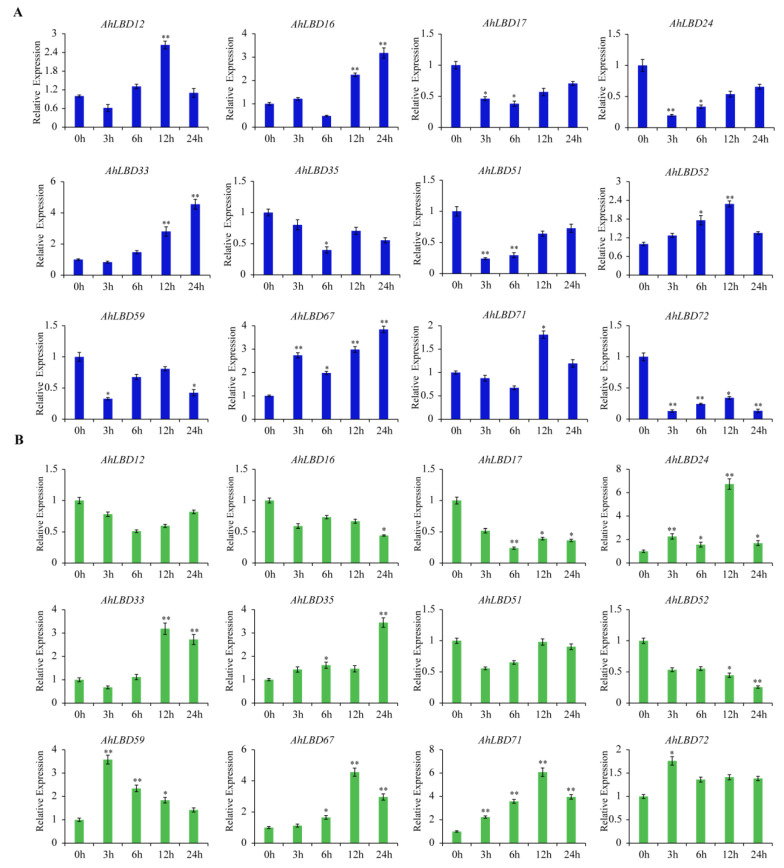
qRT-PCR results of 12 *AhLBD*s under salt and drought treatments. (**A**) 12 *AhLBD* gene expression levels in root under 2% NaCl treatments. (**B**) 12 *AhLBD* gene expression levels in root under 30% PEG6000 treatments. Error bars indicate standard deviations among three independent biological replications. 0 h was used as the mock. *AhACT11* was used as the internal control. *t*-test was used to analyze three biological replicates of each sample. *: *p* < 0.05, **: *p* < 0.01.

**Figure 15 ijms-25-10561-f015:**
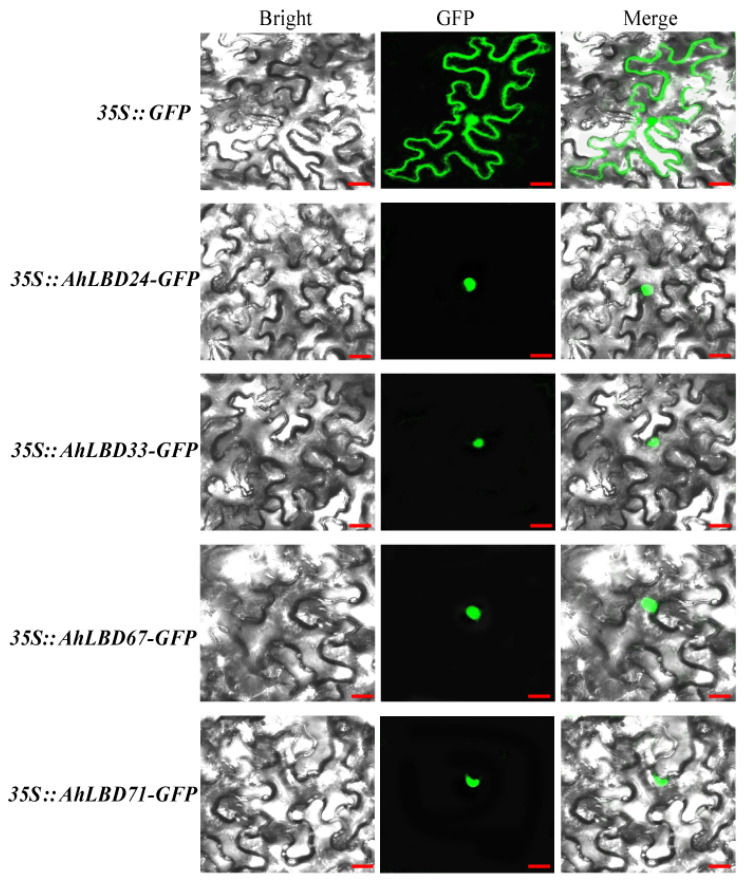
Subcellular localization of AhLBD24, AhLBD33, AhLBD67, and AhLBD71 proteins in tobacco cells. GFP driven by the CaMV 35S promoter was used as a control. Green fluorescence was observed 48–72 h post-Agrobacterium infiltration using confocal microscopy (Bars = 50 μm).

**Figure 16 ijms-25-10561-f016:**
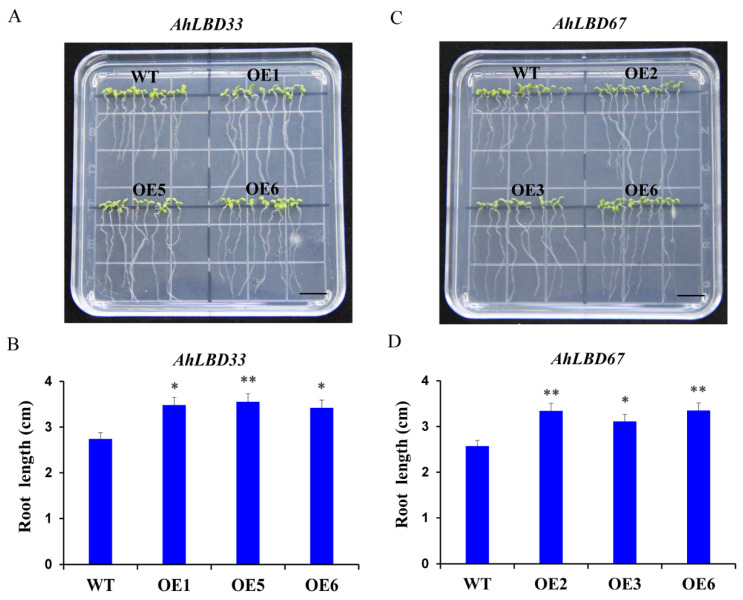
Salt stress identification of *AhLBD33* and *AhLBD67* overexpression in *Arabidopsis*. (**A**) and (**B**) Comparison of root length between *AhLBD33 OE* lines and WT. (**C**) and (**D**) Comparison of root length between *AhLBD67 OE* lines and WT. OE: overexpression line, Bar = 1 cm. *: *p* < 0.05, **: *p* < 0.01.

## Data Availability

The data presented in this study are available in the article.

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
