# Peer review of "Genome-Wide Analysis Elucidates the Roles of AhLBD Genes in Different Abiotic Stresses and Growth and Development Stages in the Peanut (Arachis hypogea L.)"

_ijms, 2024, doi:10.3390/ijms251910561_

Round 1
Reviewer 1 Report
Comments and Suggestions for Authors
This manuscript described well about LBDs in peanut. All the data from Figure1~12 is about in silico genome structure analysis. Actual data is from figure 13~16. The result is very good and matched well with the in silico data. In my opinion, even if it is published in its current state, it will provide very good information to readers.
This manuscript analyzed the LBD gene in peanut at the whole genome level. In particular, in figures 11 and 12, the fact that this gene is very deeply involved in salt stress and drought stress was summarized through gene expression profiling, and this was verified through actual qRT-PCR (Figure 14). In addition, based on this data, the AhLBD33 and AhLBD67 genes were heterologously expressed and the expression of these genes demonstrated tolerance to salt stress, thereby well proving that the AhLBD gene has a resistance function to abiotic stresses.
Author Response
Thank you for your recognition of our work
Reviewer 2 Report
Comments and Suggestions for Authors
Authors study lateral organ boundaries domain in peanuts. The work shown seems comprehensive, involving gene expression, and study of the response towards abiotic stress, and they are of interest for the research community. As such, I would recommend acceptance after some comments:
- I could not find the supplementary materials within the manuscript. Authors should ensure that they are incorporated in the Final version.
- Can authors provide more information about how potential interactions between AhLBD proteins and other proteins of peanut were predicted?
- Authors collected transcriptome data with twenty-two tissue and organ samples at different developmental stages of A. hypogaea (Tifrunner). Please include more details on how this data was collected, including the developmental stages
- Authors also tested the use of PEG (20% PEG6000) stress. They should include more details about the relevance of this abiotic stressor, and why it is relevant for this study.
- There are some typos in the manuscript including:
o “And AhLBD24/59/67/72 genes had significantly higher expression in stem…”
o “his study further confirmed through molecular genetic experiments that heterologous overexpression of AhLBD33…”
Comments on the Quality of English Language
Some typos have been identified
Author Response
Authors study lateral organ boundaries domain in peanuts. The work shown seems comprehensive, involving gene expression, and study of the response towards abiotic stress, and they are of interest for the research community. As such, I would recommend acceptance after some comments:
I could not find the supplementary materials within the manuscript. Authors should ensure that they are incorporated in the Final version.
Answer 1: Thanks for your suggestion. I will upload the supplementary materials.
Can authors provide more information about how potential interactions between AhLBD proteins and other proteins of peanut were predicted?
Answer 2: The AhLBD protein sequences were uploaded to the STRING database (https://stringdb.org/) for node comparison, and relationships among important proteins were predicted based on the Arabidopsis protein interactions data. Cytoscape (V3.7.1) was used to visualize the resulting network. The file of go-basic.obo was downloaded from the website (http://purl.obolibrary.org/obo/go/go-basic.obo/). GO anno-tation was analysed by TBtools and visualized by using the online software tool (https://www.bioinformatics.com.cn/) .
Authors collected transcriptome data with twenty-two tissue and organ samples at different developmental stages of A. hypogaea (Tifrunner). Please include more details on how this data was collected, including the developmental stages
Answer 3: There are 21 tissue organs in the heat map in this study (Fig.11), so there is an error here, which we have corrected in the manuscript. The RNA-seq data of 21 distinct tissue types and developmental stages (accession: PRJNA291488), along with cold, heat, salt and PEG treatments (accession: PRJNA553073) were downloaded from NCBI database (https://www.ncbi.nlm.nih.gov/). All gene expression levels were normalized by log2(FPKM+1).
Authors also tested the use of PEG (20% PEG6000) stress. They should include more details about the relevance of this abiotic stressor, and why it is relevant for this study.
Answer 4: In this study, cis-element analysis showed that there were 19 AhLBDs contained drought-induced functional elements in the peanut AhLBD gene family(Fig.6). Further qRT-PCR identification of selected 12 AhLBD genes was performed through drought stress treatment, confirming that the peanut LBD family responds to drought stress. Besides, LBD family genes have also been studied in drought stress in other crops. For example, the expression of GmLBD12 could be induced by drought, salt, cold, and hormones [1]. Expression of StLBD2-6 and StLBD3-5 in the stem was induced under drought stress in potato [2].
- Yang, H.; Shi, G.; Du, H.; Wang, H.; Zhang, Z.; Hu, D.; Wang, J.; Huang, F.; Yu, D. Genome-Wide Analysis of Soybean LAT-ERAL ORGAN BOUNDARIES Domain-Containing Genes: A Functional Investigation of GmLBD12. Plant Genome 2017, 10, 10.3835.
- Liu, H.; Cao, M.; Chen, X.; Ye, M.; Zhao, P.; Nan, Y.; Li, W.; Zhang, C.; Kong, L.; Kong, N.; et al. Genome-Wide Analysis of the Lateral Organ Boundaries Domain (LBD) Gene Family in Solanum tuberosum. Int. J. Mol. Sci. 2019, 20, 5360.
There are some typos in the manuscript including:
o “And AhLBD24/59/67/72 genes had significantly higher expression in stem…”
“his study further confirmed through molecular genetic experiments that heterologous overexpression of AhLBD33…”
Answer 5: Thank you for your comments. I have made changes in the manuscript.